# DNA Damage- But Not Enzalutamide-Induced Senescence in Prostate Cancer Promotes Senolytic Bcl-xL Inhibitor Sensitivity

**DOI:** 10.3390/cells9071593

**Published:** 2020-07-01

**Authors:** Nicolas Malaquin, Arthur Vancayseele, Sophie Gilbert, Laureen Antenor-Habazac, Marc-Alexandre Olivier, Zakia Ait Ali Brahem, Fred Saad, Guila Delouya, Francis Rodier

**Affiliations:** 1Francis Rodier Lab, Institut du Cancer de Montréal, Centre de Recherche du Centre Hospitalier de L’université de Montréal (CRCHUM), Montreal, QC H2X 0A9, Canada; nico.malaquin@gmail.com (N.M.); arthur.vancayseele@gmail.com (A.V.); ahlaureen@gmail.com (L.A.-H.); oliviermarcalexandre@gmail.com (M.-A.O.); zakia.ait.ali.brahem@umontreal.ca (Z.A.A.B.); g.delouya@gmail.com (G.D.); 2Fred Saad Lab, Institut du Cancer de Montréal, Centre de Recherche du Centre Hospitalier de L’université de Montréal (CRCHUM), Montreal, QC H2X 0A9, Canada; sophiegilbe@gmail.com (S.G.); fred.saad@umontreal.ca (F.S.); 3Département de Chirurgie, Université de Montréal, Montreal, QC H3C 3J7, Canada; 4Département de Radiologie, Radio-Oncologie et Médecine Nucléaire, Université de Montréal, Montreal, QC H3T 1A4, Canada

**Keywords:** prostate cancer, cellular senescence, senolytics, PARP inhibitor, enzalutamide

## Abstract

Cellular senescence is a natural tumor suppression mechanism defined by a stable proliferation arrest. In the context of cancer treatment, cancer cell therapy-induced senescence (TIS) is emerging as an omnipresent cell fate decision that can be pharmacologically targeted at the molecular level to enhance the beneficial aspects of senescence. In prostate cancer (PCa), TIS has been reported using multiple different model systems, and a more systematic analysis would be useful to identify relevant senescence manipulation molecular targets. Here we show that a spectrum of PCa senescence phenotypes can be induced by clinically relevant therapies. We found that DNA damage inducers like irradiation and poly (ADP-ribose) polymerase1 (PARP) inhibitors triggered a stable PCa-TIS independent of the p53 status. On the other hand, enzalutamide triggered a reversible senescence-like state that lacked evidence of cell death or DNA damage. Using a small senolytic drug panel, we found that senescence inducers dictated senolytic sensitivity. While Bcl-2 family anti-apoptotic inhibitor were lethal for PCa-TIS cells harboring evidence of DNA damage, they were ineffective against enzalutamide-TIS cells. Interestingly, piperlongumine, which was described as a senolytic, acted as a senomorphic to enhance enzalutamide-TIS proliferation arrest without promoting cell death. Overall, our results suggest that TIS phenotypic hallmarks need to be evaluated in a context-dependent manner because they can vary with senescence inducers, even within identical cancer cell populations. Defining this context-dependent spectrum of senescence phenotypes is key to determining subsequent molecular strategies that target senescent cancer cells.

## 1. Introduction

Most prostate cancer (PCa) cases respond to first-line treatments including surgery, irradiation (XRA), and androgen deprivation therapy (ADT), but 10% will progress to metastatic castrate-resistant PCa (mCRPC) [1]. Combination radio-ADT therapies have shown improvement over single agents, suggesting that new clinical strategies can improve the efficacy of current PCa therapies [2,3,4]. PCa growth is stimulated by androgen receptor (AR) signaling pathways that can be directly inhibited by AR antagonists called anti-androgens (AA) [5]. Among new targeted therapies, Enzalutamide (Enza) is an emerging AA that was FDA-approved for patients with mCRPC [6,7,8]. Similarly, PARP inhibitors (PARPi) represent a new class of PCa anticancer agents that target cancer cells harboring DNA repair defects, including *BRCA* or *ATM* mutations [9]. PARPi olaparib (Olap) and rucaparib recently received FDA-breakthrough designations for *BRCA1/2*-mutated mCRPC [10]. However, not all patients harboring *BRCA* mutations respond well to PARPis, and their clinical use as maintenance monotherapy in ovarian cancer gives rise to resistance, suggesting a similar risk for PCa [11,12]. Therefore, understanding the cellular responses behind current PCa therapies will improve our mechanistic knowledge to identify molecular targets and improve the efficiency of emerging treatments.

Cellular senescence is a multifaceted stress response involved in tumor suppression, tissue repair, aging, as well as cancer therapy [13,14,15,16]. Key SA phenotypic hallmarks include SA-β-galactosidase (SA-β-gal) activity, persistent DNA damage response (DDR) activation; a proinflammatory secretory phenotype (SASP) constituted of cytokines (i.e., IL-6 and IL-8), growth factors and proteases; and apoptosis resistance (SAAR) through an upregulation of the Bcl-2 antiapoptotic protein family [13,17,18,19,20,21,22,23]. At its core, senescence is defined by a stable senescence-associated proliferation arrest (SAPA) governed by two major tumor suppressor pathways, p53/p21^Cip1^ and p16^INK4a^/Rb [24,25,26]. Despite high p16^INK4a^ or p53 mutation rates, multiple evidences show that cancer cells can retain the capacity to develop some senescence-associated (SA) phenotypes in response to treatment (Therapy-induced senescence or TIS) [16,20,27,28,29,30,31]. Most localized (non-aggressive) PCa retain normal p53 status, suggesting that human prostate cells bypass the natural tumor suppression aspect of senescence without losing p53 functions. Alternatively, aggressive PCa almost always lack p53 functions [32]. Independent of p53 status, PCa cells can undergo TIS in response to radiotherapy and DNA-damaging chemotherapies [20,33,34,35,36] including PARPis [37,38], charcoal-mediated ADT [18] and Enza treatment [39,40,41].

Because the stability of the TIS proliferative arrest can be weakened by the high rates of p53 or p16 mutations in cancer cells including PCa, senescence reinforcement or manipulation strategies could reduce the risk of cancer recurrence [31,42]. Also, TIS cells that persist in tissues can create a microenvironmental niche suitable for tumor resistance [16,17,43,44,45,46], overall suggesting that the elimination of TIS cells may improve the outcome of cancer therapy. We and others have developed a “one-two punch” strategy which selectively targets TIS cells using senolytics drugs [31,47,48]. Many senolytics (i.e., piperlongumine (PPL), fisetin, quercetin + dasatinib) are efficient in improving healthy lifespan and slowing age-related diseases progression in vivo [49,50]. In the context of high-grade serous ovarian cancer and triple-negative breast cancer, we previously demonstrated that PARPi-TIS cells were particularly sensitive to Bcl-2/Bcl-xL inhibitors, including ABT-263, which triggered PARPi-TIS cells senolysis and consequently improved treatment outcomes in vitro and in vivo [31,51].

Although some therapies can trigger TIS in PCa, the SA molecular and cellular characteristics may differ depending on the treatment. It remains unclear if all types of TIS can be targeted by senolytics or manipulated in different ways for example to reinforce the senescence proliferation arrest. Here, we characterized TIS in PCa cells treated with XRA, Olap or Enza and investigated whether PCa-TIS can be eliminated using senolytics to re-direct senescent cells towards apoptosis. Using LNCaP and PC-3 cell lines respectively representing prostatic castrate-sensitive adenocarcinoma and castrate-resistant small cell neuroendocrine carcinoma (SCNC) metastatic cells [52], we found that XRA- and Olap-TIS cells were targetable using Bcl-2 family inhibitors while Enza-TIS cells resisted such senolysis. Interestingly, the previously described senolytic PPL acted to reinforce Enza-TIS proliferation arrest without triggering cell death. This suggests that multiple layers of PCa-TIS manipulation may advance new treatment strategies for mCRPC when used in pre-defined contexts.

## 2. Materials and Methods

### 2.1. Cells and Culture Conditions

PCa cell lines PC-3 and LNCaP given by Dr. Fred Saad’s laboratory (CRCHUM) were cultured in RPMI (350-000-CL, Wisent, Saint-Jean-Baptiste, QC, Canada) supplemented with 10% FBS (12483, Gibco, Thermo Fisher, Waltham, MA, USA), 100 IU/mL penicillin and 100 µg/mL streptomycin (450-201-EL, Wisent, Saint-Jean-Baptiste, QC, Canada), and maintained at 37 °C in 20% O_2_ and 5% CO_2_ conditions.

### 2.2. Drugs

Olaparib/Olap (AZD2281) and A-1155463/A-115 (S7800) were purchased from Selleckchem, Houston, TX, USA. ABT-263 (Navitoclax, A3007) and enzalutamide/Enza (MDV3100, A3003) were from APExBIO, Houston, TX, USA. Drugs were dissolved in 100% dimethyl sulfoxide (DMSO) and diluted in complete culture media for in vitro experiments. Drugs were added 48 h after seeding and Enza treatment was renewed every 3 days for all experiments. For long-term Enza treatments, cells were cultured in RPMI media supplemented with 10 µM Enza for 12, 18, 24 or 30 days in T175 flasks and transferred to 6-, 24- or 96-well plates for different experiments.

### 2.3. Irradiation

Cells were X-irradiated with a total dose of 8 Gy at rates equal to or above 0.75 Gy/min using Gammacell^®^ 3000 irradiator Elan.

### 2.4. IncuCyte^®^ Cell Proliferation Phase-Contrast Imaging Assay

To evaluate cell proliferation, 1500 cells per well were seeded in 96-well plates (all cells expressing H2B-GFP). Cells were incubated with PARPi, Enza and senolytic drugs in adjusted 0.75% DMSO. Cell number was imaged by phase contrast and fluorescence using the IncuCyte^®^ Live-Cell Imaging System (Essen BioScience Inc., Ann Arbor, MI, IncuCyte zoom). Frames were captured at 6-h intervals in three independent wells using a 10× objective. Proliferation growth curves were constructed using IncuCyte^®^ Zoom software (Essen BioScience Inc., Ann Arbor, MI, USA, V2019B) from measurements of H2B-GFP cell nuclei numbers. Each experiment was performed in triplicate and repeated three times.

### 2.5. SA-β-galactosidase Detection

SA-β-gal assay was performed as previously described [53]. Briefly, cells grown in 6-well plates were washed with 1× PBS, fixed with 10% formalin for 5 min, washed again with PBS and incubated at 37 °C for 16 h in a staining solution composed of 1 mg/mL 5-bromo-4-chloro-3-inolyl-β-galactosidase in dimethylformamide (20 mg/mL stock), 5 mM potassium ferricyanide, 150 mM NaCl, 40 mM citric acid/sodium phosphate and 2 mM MgCl_2_, at pH 6.0. Cells were washed with PBS, fixed with 10% formalin for 10 min, and washed again with PBS. Pictures were taken for quantification.

### 2.6. Immunofluorescence

LNCaP and PC-3 cells were seeded onto coverslips in 24-well plates, treated 48 h after seeding, and grown for 6 days. Cells were fixed in formalin for 10 min at room temperature (RT) and permeabilized in 0.25% Triton in PBS for 15 min. Slides were blocked for 1 h in PBS containing 1% BSA and 4% donkey serum. Primary antibodies (53BP1, Novus, Saint Charles, MI, USA, NB100-305; γH2AX, Millipore, Burlington, MA, USA, JBW301) diluted in blocking buffer were added and slides were incubated overnight at 4 °C. Cells were washed and incubated with secondary antibodies for 1 h at RT, then washed again. Coverslips were mounted onto slides using Prolong^®^ Gold anti-fade reagent with DAPI (Life Technologies Inc., Carlsbad, CA, USA). Images (400× magnification) were obtained using a Zeiss microscope AxioObserver Z1, Carl Zeiss, Jena, Germany. Automated analysis AxioVision™ software from Zeiss, Jena, Germany (AxioVision SE64 Rel. 4.9.1) was used to count foci to calculate the average number of foci per nucleus. The fold change was calculated as the ratio between percentages of γH2AX or 53BP1 nuclear foci in treated versus control (non-treated) cells. γH2AX and 53BP1 foci were quantified in > 150 nuclei from three different fields of each coverslip.

### 2.7. Cell Cycle and Cell Death Analysis by Flow Cytometry

LNCaP and PC-3 cells were seeded in 6-well plates and treated 48 h after seeding, then harvested 6 days after. For cell cycle analysis, live cells were fixed for 24 h in 70% ethanol and incubated for 30 min at RT with 100 μg/mL RNAse A and 25 μg/mL propidium iodide. For cell death analysis, cells were incubated 5 min at RT with DRAQ7 (ab109202, Abcam Inc., Cambridge, MA, USA). A maximum of 30,000 events were counted per condition using the Fortessa flow cytometer (BD Biosciences, Mississauga, ON, Canada) and analyzed with FlowJo software (Ashland, OR, USA, V10).

### 2.8. EdU (5-ethynyl-2′-deoxyuridine) Detection

To detect DNA synthesis, cells were seeded onto coverslips in 24-well plates and treated 24 h after seeding. EdU (10 μM, Invitrogen, Carlsbad, CA, USA) was added to the medium and incubated for 24 h before the end of drug treatment on days 2 or 6. Cells were washed three times with TBS and fixed with 10% formalin for 10 min. EdU staining was assessed using the Click-iT^®^ EdU Alexa Fluor^®^ 647 Imaging Kit (Invitrogen, Carlsbad, CA, USA). Coverslips were mounted onto slides using Prolong^®^ Gold anti-fade reagent with DAPI (Life Technologies Inc., Carlsbad, CA, USA). Images were obtained using a Zeiss microscope and automated analysis software from Zeiss was used to count foci.

### 2.9. Drug Combination Analysis

LNCaP and PC-3 cells were infected with lentiviruses encoding H2B-GFP and treated with the different combination treatments. Cell number was imaged by phase contrast and fluorescence using the IncuCyte^®^ Live-Cell Imaging System (IncuCyte HD). For each concentration of senolytic drug, the senolytic index (SI) was calculated as follows: SI = [(Nf/Ni) SX/(Nf/Ni) S0] No PARPi/[(Nf/Ni) SX/(Nf /Ni) S0] PARPi, where Nf = Final cell number; Ni = initial cell number; SX = X concentration of the senolytic; and S0 = no senolytic.

### 2.10. Cloning, Viruses and Infections

Viruses were produced as described previously and titers were adjusted to achieve ~90% infectivity [13]. Infections were followed 48 h later by puromycin or hygromycin selection and stable cells were either used or stored at −80 °C. Lentiviruses encoding H2B-GFP were produced by amplification of the H2B sequence from the pENTR1A-H2B-HcRed plasmid (a gift from Dr. Richard Bertrand, CRCHUM, Canada) using the following primers: ES-92 (5′-GGTACCCCACCATGCCAGAGCCAGCGAAGTCTGCT-3′) and ES-93 (5′-GGATCCTAGCGCTGGTGTACTTGGTGATGG-3′). The amplified product was digested with restriction enzymes Kpn1/BamH1 and inserted into the entry vector pENTR1A-GFP-N2 (FR1) to generate pENTR1A-H2B-GFP, which was recombined into the destination vector (pLenti PGK Hygro Dest (W530-1)) to obtain the final lenti-H2B-GFPhygro lentivector.

### 2.11. Real-Time Quantitative Polymerase Chain Reaction (Q-PCR)

Total RNA was extracted from harvested cells using the RNeasy kit (Qiagen Inc., Hilden, Germany). One microgram of total RNA was subjected to reverse transcription using the QuantiTect Reverse Transcription Kit (Qiagen Inc., Hilden, Germany). One microliter of the reverse-transcribed product was diluted (1:10) and subjected to Q-PCR using sequence specific primers (400 nM) and the SYBR Select Master Mix (Applied Biosystems^®^, Life Technologies Inc., Carlsbad, CA, USA). Q-PCR was performed using Applied BioSystems^®^ Step One Plus apparatus (UDG activation 50 °C/2 min, followed by AmpliTaq activation plus denaturation cycle 95 °C/2 min, followed by 40 cycles at 95 °C/15 s, 60 °C/1 min and 72 °C/30 s). Gene expression values were normalized to TATA-binding protein gene expression. Three independent experiments were performed in duplicate. Sequence primers for target genes are described in Appendix A.

### 2.12. Preparation of Conditioned Media and Analysis of Secreted SASP Factors

Treated cells were seeded in 6-well plates (100,000 cells per well) and irradiated 24 h after seeding. Untreated cells were similarly seeded 48 h before the preparation of conditioned media as previously described [54]. Briefly, cells were washed three times with serum-free medium 8 days after irradiation, followed by incubation in serum-free RPMI medium containing antibiotics for 16 h. Cells were manually counted and conditioned media was stored at −80 °C until assayed. Conditioned media was analyzed using the multispot electrochemiluminescence immunoassay system for 40 secreted factors using the V-Plex human kit from Meso Scale Discovery (MSD, Rockville, MD, USA; #K15209D) following the manufacturer’s instructions as performed previously [31]. The results are reported as concentrations of target protein normalized to cell number.

## 3. Results

### 3.1. Irradiation and Olaparib Trigger Senescence-Associated Phenotype in LNCaP Cells

To investigate the impact of XRA and Olap treatment on PCa cell fate decisions, we selected LNCaP, a metastatic, androgen-dependent, p53 wild-type PCa cell line that is characteristic of human prostatic adenocarcinoma [52,55]. LNCaP proliferation was arrested by XRA (Figure 1A, Appendix A) or reduced by Olap treatment in a dose-response relationship (Figure 1B, Appendix A). Less than 10% cumulative cell death was observed by flow cytometry 6 days after both treatments (Figure 1C, Appendix A). DNA content assessed by propidium iodide showed that the G2 cell population of LNCaP was strongly increased and less than 1% of cells were in S-phase (Figure 1D, Appendix A). Moreover, less than 5% of cells had aneuploidy (8N) (Figure 1D, Appendix A) suggesting that XRA or Olap triggered only mild genomic instability. Within 24 h of treatment, 80% of cells were already negative for DNA synthesis and remained negative for up to 6 days post-treatment, demonstrating stable proliferation arrest (Figure 1E, Appendix A). Furthermore, most LNCaP cells were positive for SA-β-gal activity suggesting that XRA and Olap triggered senescence in LNCaP (Figure 1F, Appendix A). These senescent cells displayed persistent DNA double strand-breaks damage foci labelled by 53BP1 and phosphorylated histone H2AX (γH2AX) [56] (Figure 1G), and increased expression of SA genes, including the cyclin-dependent kinase inhibitors (CDKi) p21, p16 and p15 (Figure 1H). Expression of typical SASP components such as IL-6, IL-8 or IL-1α were not significantly increased following XRA or Olap treatments except for IL-1β (Figure 1I). A multiplex assay showed that the secretion of only a few SASP factors like VEGF, MCP-1 or IL-8 were increased in the conditioned media of irradiated LNCaP (Appendix A). Altogether these data show that XRA and Olap mainly trigger a DNA damage-induced senescence phenotype in LNCaP cells.

### 3.2. Irradiation and Olaparib Trigger Senescence, Cell Death and Mitotic Catastrophe in PC-3 Cells

To investigate the cell fate decisions of more aggressive PCa cells in response to radiotherapy and Olap, we selected PC-3, a metastatic, p53-mutated PCa cell line that harbours characteristics of prostatic SCNC [52]. Similar to LNCaP cells, we observed stable proliferation arrest of PC-3 cells in response to XRA (Figure 2A, Appendix A) and dose-response reduction of proliferation induced by Olap, although the Olap response was slower compared to LNCaP (Figure 2B, Appendix A; Figure 1B, Appendix A). In contrast to LNCaP, a mix of living attached cells and floating dead cells (about 30% cumulative) were observed 6 days after both treatments (Figure 2C, Appendix A). The remaining living cells were mostly accumulated in G2 phase with practically no cells in S-phase (Figure 2D, Appendix A). Importantly, about 20% of the treated PC-3 cells were aneuploid (8N) (Figure 2D, Appendix A), suggesting that a mitotic catastrophe occurred in these cells. Indeed, some characteristics of mitotic catastrophe including DNA synthesis (EdU-positive cells) 2 days post-treatment (Figure 2E, Appendix A) and abnormal nucleus content (multiple nuclei or micro-nuclei) (Figure 2G, Appendix A) were observed in XRA- and Olap-treated PC-3 cells. In addition to cell death, mitotic catastrophe can also promote senescence [57]. At 6 days post-treatment, 90% of living PC-3 cells were negative for EdU-pulse labeling, demonstrating stable proliferation arrest (Figure 2E, Appendix A). About 30% of the cells were positive for SA-β-gal activity (Figure 2F, Appendix A) and most displayed persistent DNA damage foci (Figure 2G). Finally, we observed an over-expression of SA genes in treated PC-3 cells, including CDKi (Figure 2H) and several SASP factors (Figure 2I). As we have previously seen for other types of senescent cancer cells [31], a high level of many secreted SASP factors including IL-6 and IL-8 was detected in the conditioned media of irradiated PC-3 cells by multiplex assay, suggesting that the low-level SASP observed in LNCaP cells is not related to prostate cancer in general (Appendix A). Hence, DNA damage treatments XRA and Olap trigger a mitotic catastrophe process in the SCNC PC-3 cell line, resulting in genomic instability, cell death and a high proportion of senescent cells.

### 3.3. Combination of Irradiation or PARPi Treatments with Senolytics

We investigated whether XRA- and Olap-induced TIS cells could be eliminated using a small panel of senolytic drugs. We selected two inhibitors of the Bcl-2 family: ABT-263 (navitoclax), which inhibits both Bcl-2 anti-apoptotic family members Bcl-2 and Bcl-xL; and A-1155463 (A-115), which targets selectively Bcl-xL. We also selected piperlongumine (PPL), a natural product that selectively kills senescent cells including senescent ovarian cancer cells [31,49]. To highlight the senolytic sensitivity of PCa-TIS cells, we used a single dose of XRA (8 Gy) or Olap (5 µM) in LNCaP and PC-3 to induce an optimal response of senescence in combination with increasing concentrations of ABT-263, A-115 or PPL (concentrations given in Appendix A). For each combination, surviving cells were counted after 6 days of treatment (Figure 3A,B, Appendix A) and the synergistic effect of the combination was evaluated using the Bliss independent model [58] in which negative values indicate antagonism, values around zero indicate additive effects and positive values indicate synergy (Appendix A). Interestingly, most of the ABT-263 and A-115 concentrations worked in synergy with XRA and Olap to reduce LNCaP and PC-3 cell numbers, but not PPL (Appendix A). To determine whether ABT-263, A-115 or PPL selectively targeted XRA- or Olap-TIS in LNCaP and PC-3 cells, we compared the number of surviving cells in response to senolytics for each senescence inducer (Figure 3A,B, Appendix A). We also calculated a senolytic index (SI) as we previously described [31]. SI is based on the ratios of surviving cells (over control) for each combination in order to reflect the capacity of senolytic drugs to specifically eliminate senescent cells (Figure 3C,D). The SI for most of the ABT-263 and A-115 concentrations, but not for PPL, were elevated in LNCaP and PC-3 cells (Figure 3C,D). Finally, to confirm that ABT-263 and A-115 induced cell death in XRA- or Olap-TIS cells, we analyzed cumulative cell death after 6 days of treatments (Figure 3E,F, Appendix A). The combination of 0.625 µM ABT-263 or 0.3125 µM A-115 with XRA or Olap increased cell death up to 80% in PC-3 and LNCaP cells (Figure 3E,F, Appendix A). Altogether these data show that the Bcl-2 family inhibitors (ABT-263 and A-115) but not PPL can be used as senolytics to trigger cell death in XRA- or Olap-TIS PC-3 and LNCaP cells.

### 3.4. Enza Triggers a Senescence-Like State in LNCaP Cells that is Resistant to Bcl-2 Family Senolytics

Enza is a second-generation AA treatment for both non-metastatic PCa and mCRPC. Enza directly binds to the AR to inhibit its activity and prevent tumor growth [59,60,61]. Although Enza can trigger senescence in LNCaP cells [39,40,41], the characterization of Enza-TIS senescence hallmarks remains poorly described. We treated LNCaP (androgen-sensitive) cells with increasing concentrations of Enza and analyzed the impact on cell proliferation using live-cell imaging. We found that 10 µM of Enza was sufficient to effectively block LNCaP proliferation (Figure 4A, Appendix A) but contrary to XRA or Olap, Enza did not trigger any detectable cell death in LNCaP (Figure 4B, Appendix A). The vast majority of Enza-treated cells were enriched in the G1 cell cycle phase after 12 days of treatment with complete absence of abnormal DNA content (8N), suggesting that Enza did not induce mitotic catastrophe in LNCaP cells (Figure 4C,D, Appendix A). Importantly, the percentage of SA-β-gal positive cells gradually increased over time with Enza exposure, suggesting a slow induction of senescence (Figure 4E, Appendix A) but contrary to XRA or Olap treatments (Figure 1G and Figure 2G), Enza did not increase the number of persistent DNA damage foci even after 12 days of treatment (Figure 4F). We observed that Enza increased the expression of p15, p16 and IL-1β (Figure 4G) but not of p21 as in XRA- or Olap-TIS cells (Figure 1H and Figure 2H). These results suggest that Enza triggered a senescence-like phenotype without DNA-damage in LNCaP.

We then investigated whether Enza-TIS LNCaP cells could be targeted with senolytics. We treated LNCaP cells with 10 µM Enza for 6 or 12 days, in combination with increasing concentrations of ABT-263, A-115 or PPL for the last 6 days (concentrations in Appendix A). For each combination, surviving cells were counted (Appendix A) and the synergistic effect of the combination was evaluated by calculating a Bliss score (Figure 5A). PPL but not ABT-263 or A-115 synergistically decreased the number of cells treated with Enza for 6 days (Figure 5A). Intriguingly, this synergetic effect of PPL was partially lost after 12 days of combinatory treatment (Figure 5A). We also compared the effect of the senolytics on LNCaP cells by evaluating the percentage of cell survival (Figure 5B) and by calculating the SI (Figure 5C). Except for the highest dose of PPL at day 6, most of the combinations revealed low SI synergy, suggesting PPL acted on cell proliferation rather than survival (Figure 5C). Consistently, we did not observe significantly increased cell death in Enza-TIS cells following ABT-263, A-115 or PPL treatments (Figure 5D). This demonstrates that Enza-senescent LNCaP cells are not sensitive to inhibitors of the Bcl-2 antiapoptotic family but that PPL may act as a senomorphic or seno-modulator by reinforcing the Enza-induced proliferation arrest.

### 3.5. Senescence-Like State Induced by Enzalutamide is Reversible

Since senescence phenotypes can develop over extended periods [10,62,63], we examined if long-term Enza treatment could lead to DNA damage and ABT-263 or A-115 sensitivity. We exposed LNCaP cells to 10 µM Enza for 18, 24 or 30 days and evaluated the evolution of SA markers by performing cell fate characterization experiments. Compared to 12 days’ Enza-treated LNCaP, the percentage of SA-β-gal positive cells (about 30%; Figure 6A and Appendix A) and EdU-positive cells (under 20%; Figure 6B and Appendix A) remained stable. We did not observe any significant changes in the number of 53BP1 and γH2AX foci between the different time points, which suggests that the phenotypes observed at 12 days did not evolve over time (Appendix A). In contrast to XRA treatments for which a single dose led to stable proliferation arrest for at least 6 days, the release of Enza treatment for 6 days after 12, 18 or 24 days treatments led to resumption of DNA synthesis, nearly to the same level of untreated control cells (Figure 6C). This suggests that the senescence-like state induced by Enza is reversible. Accordingly, long-term Enza-treated LNCaP cells resumed their proliferation when Enza was released for 9 days after 12, 18, 24 or 30 days of treatment, albeit longer treatments required a longer recovery (Figure 6D,E). We also observed that increasing concentrations of ABT-263, A-115 or PPL for the last 6 days of the 18, 24 and 30 days of Enza treatment did not lead to a synergistic decrease of cell viability (Figure 6F,G). On the contrary, long-term Enza-treated LNCaP cells were more resistant to senolytic concentrations that normally lead to cell death in untreated control cells (Figure 6F,G).

## 4. Discussion

Cancer is associated with the presence of mutated or defective cell cycle checkpoints that can alter therapy-induced cell fate decisions, shifting away from apoptosis or senescence and towards crisis and mitotic catastrophe [64,65]. Despite these mutations, cancer cells can often retain the capacity to enter senescence in response to treatments [16,20,29,30]. In PCa, DNA damaging chemotherapy, radiotherapy [20,33,34,35,36,37,38,66], ADT [18,67,68,69,70] or Enza [39,40,41] can induce TIS, suggesting that PCa-TIS is a relevant target to enhance treatment outcomes.

Here, we demonstrate that different PCa therapies can promote senescence phenotypes in PCa cells but with major differences in SA phenotypic characteristics. The DNA damage inducers, XRA and Olap, triggered classical TIS in PCa cells in addition to the mitotic catastrophe and cell death observed in PC-3 cells. Although mitotic catastrophe could be explained by differences in cell cycle regulation, particularly those related to the p53 status, no causal link can be established yet due to the many other genomic differences that exist between PC-3 and LNCaP [71]. LNCaP cells have increased levels of p15, p16 and p21 CDKi transcripts, but PC-3 cells only express p21. This observation could be partially explained by higher levels of CBX7, a repressor of the p16^INK4a^/p14^ARF^
*CDKN2A* locus [68,72], found in PC-3 cells. Also, regulation of p21 in the p53-mutated PC-3 cell line is mediated by FOXO proteins [73], which are implicated in senescence [74,75], PCa growth and malignancy [76,77,78,79,80], and have been shown to be upregulated following PARP inhibition [81]. Similarly, the DDR checkpoint protein Chk2 is known to promote senescence through a p53-independent activation of p21 transcription in breast carcinoma, high grade serous ovarian cancer or immortalized keratinocyte cells [31,82].

Alternatively, Enza also induced TIS in androgen-sensitive LNCaP cells in the absence of direct DNA damage. In addition to DNA damage, we show that Enza-TIS cells did not exactly share other SA markers observed in XRA- or Olap-senescent cells as they displayed: (i) a more gradual loss of DNA synthesis with an accumulation of cells in G1 phase; (ii) no more than 30% SA-β-gal positive cells; (iii) a complete absence of additional DNA damage foci or genomic instability; (iv) increased p16 but not p21 expression, and (v) a reversible proliferation arrest. Indeed, although p16 expression is often associated with an irreversible state of senescence [24,25], Enza treatment for more than 30 days sustained a senescence that was rapidly reversible upon drug removal, suggesting that Enza-TIS depends on different underlying molecular mechanisms for both induction and maintenance when compared to DNA damage-TIS. Although overall Enza-TIS was reversible, we noted that longer Enza treatment yielded incrementally slower proliferation recovery (Figure 6E), suggesting that either cells acquired altered (slower) cell cycle characteristics or an increasing proportion of cells are converted to a stable form of senescence over time. Interestingly, some reports showed that androgen depletion using charcoal-stripped serum induced senescence with persistent DNA damage foci in LNCaP cells via reactive oxygen species production [18,67,68,69,70], which contrasts to what we and others observed for Enza-TIS [40]. Charcoal-stripped androgen depletion TIS was associated with the upregulation of p27^Kip1^ via the downregulation of its proteolysis through reduced activity of the E3 ligase Skp2 [67,68]. We did not measure p27^Kip1^ protein levels but the absence of p27^Kip1^ transcripts that we observed in Enza-TIS is consistent with this transcription-independent mechanism of regulation. Although it remains unclear why charcoal-stripped androgen depletion and Enza TIS are different, we can suggest that perhaps charcoal depletes other growth-survival factors. Nevertheless, Enza is currently used in clinic and understanding Enza-TIS is particularly timely.

We and others have proposed to manipulate senescence to improve cancer treatment using a “one-two punch” strategy [30,31,83,84]. This strategy targeting TIS cancer cells with senolytics has shown success in breast, ovarian, melanoma and liver cancers [31,85]. However, senolytics have been proposed to vary in efficiency depending on cellular contexts [84]. Accordingly, we found that inhibitors of the Bcl-2 antiapoptotic family, ABT-263 and A-115, were strongly efficient to kill XRA- and Olap-TIS cells but not Enza-TIS cells [31]. The mechanism underlying this latter phenomenon remains unclear but suggests DNA damage is key for sensitization to Bcl-2 family senolytics. However, a recent study also highlights the resistance of Enza-TIS LNCaP to ABT-263, but interestingly reveals that Enza-treated LNCaP cells undergo increased apoptosis upon treatment with an allosteric Akt inhibitor (MK2206) whether or not they were senescent, suggesting this compound blocks an essential survival pathway in these cells [41]. Along the same lines, the senolytic PPL is a compound shown to trigger apoptosis and preferentially kill senescent cells induced by oncogene expression, ionizing radiation or replicative exhaustion [49], but also senescent ovarian cancer cells [31]. Here we show that PPL did not display synergistic or senolytic effect on XRA- or Olap-TIS PCa cells, but had a synergistic effect with short-term Enza treatment, although it did not induce cell death in this context. Our hypothesis is that PPL enhanced the Enza-associated proliferation arrest and therefore only affected residual proliferating cells under short-term treatment. Overall, this supports the idea that the regulation network for TIS-associated apoptosis resistance or survival signals is inducer-dependent in the PCa context and provides multiple targets for pharmacologic interventions.

In addition to being resistant to senolytic concentrations that triggered senolysis in XRA- or Olap-treated cells, long-term Enza-treated LNCaP cells were resistant to higher, non-specific doses of ABT-263 in comparison to untreated cells. These interesting findings require further investigation as they could explain tumor resistance in some clinical settings. These mechanisms of resistance could be due to the acquisition of neuroendocrine features following lineage reprogramming, a well-described phenomenon linked with the acquisition of drug resistance in clinic [86]. This may also be due to autophagy upregulation, a phenomenon already observed in LNCaP cells after AA treatments including Enza and bicalutamide [87], and described as driving the resistance of PCa cells to bicalutamide, Enza, taxanes, mitoxanthrone and radiotherapy [88].

XRA is the most common first-line therapy for PCa, whereas PARPis Olap and rucaparib have now FDA-breakthrough designations for *BRCA1/2*-mutated mCRPC [10]. One study showed that PC-3 cells are characteristic of SCNC, which, although rare and present in 0.5–2% of patients, accounts for 10–20% of PCa found in autopsies of men who died from CRPC [52,89]. Our results, therefore, constitute a solid rationale for further preclinical investigation of a PCa-TIS targeted, one-two punch approach using inhibitors of the Bcl-2 anti-apoptotic protein family. Current limitations of this strategy are risks of navitoclax (ABT-263) inducing thrombocytopenia and neutropenia [90,91] due to Bcl-2 and Bcl-xL inhibition, respectively [58,81]. However, recent studies characterized Bcl-xL inhibitors that target Bcl-xL to specific E3 ligases for degradation in tumor cells but not in platelets, which poorly express this protein [92,93]. The use of such targeted inhibitors could both prevent adverse effects and enhance the proposed PCa-TIS targeting combo-therapies at least in the context of DNA damage inducers.

## Figures and Tables

**Figure 1 cells-09-01593-f001:**
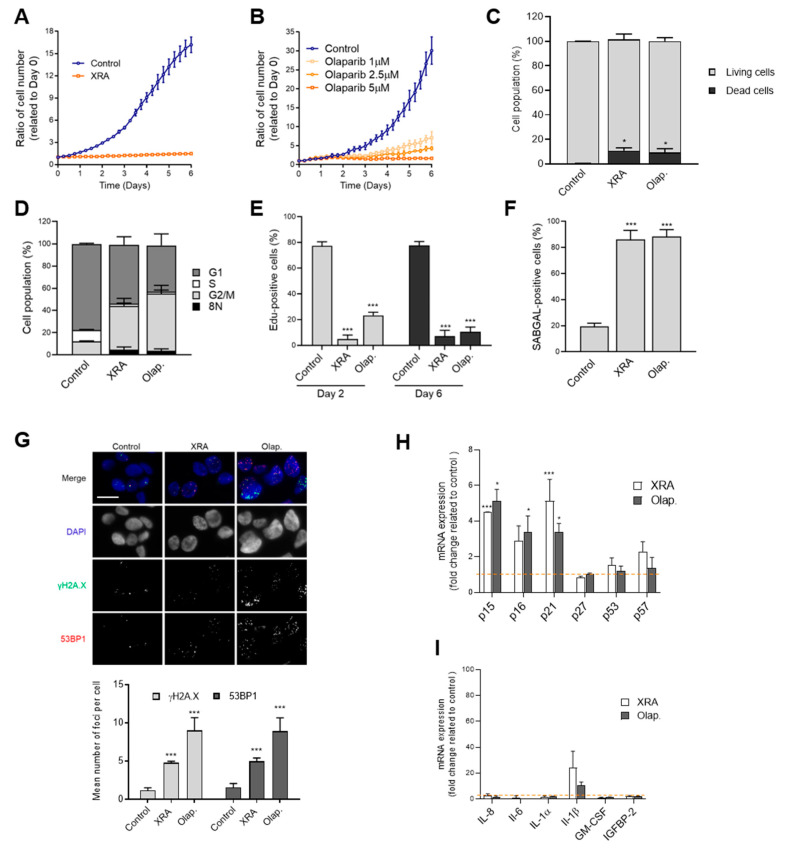
Therapy-induced DNA damage triggers senescence in LNCaP cells. (**A**,**B**) Cell proliferation curves of LNCaP expressing H2B-GFP and (**A**) irradiated with 8 Gy of X-Rays (XRA) or (**B**) treated with increasing concentrations of olaparib (Olap). Data are the mean ± SD of triplicate and are representatives of three independent experiments. (CI) LNCaP cells were irradiated with 8 Gy XRA or treated with 5 µM Olap. (**C**) The cumulative cell death of LNCaP was analyzed by flow cytometry 6 days after XRA or Olap treatment. (**D**) Flow cytometry analysis of cell cycle populations following 6 days of exposure of LNCaP cells to XRA or Olap. (**E**) Cell proliferation was assessed by incorporation of EdU for 24 h at 1 or 5 days after XRA or Olap treatment. (**F**) SA-β-gal assay was performed on LNCaP cells 6 days after XRA or Olap treatment. (**G**) Representative images (top) and quantifications (bottom) of γH2AX (green) and 53BP1 (red) foci per nucleus in LNCaP cells 6 days following XRA or Olap treatment. The merged red and green channels show colocalization in yellow and DAPI is shown in blue. Scale bar = 20 µM (**H**,**I**) Relative mRNA levels of (**H**) CDKi or (**I**) SASP factors were evaluated by real-time Q-PCR in LNCaP cells irradiated or treated with Olap for 6 days. The values represent the fold change expression related to non-treated controls (represented by the dashed line). (**C**–**I**) The mean ± SD of three independent experiments is shown. Statistical analysis: (**C**,**E**,**F**,**G**) two-tailed Student’s *t*-test; (**H**,**I**) two-way ANOVA. * *p* < 0.05, *** *p* < 0.001.

**Figure 2 cells-09-01593-f002:**
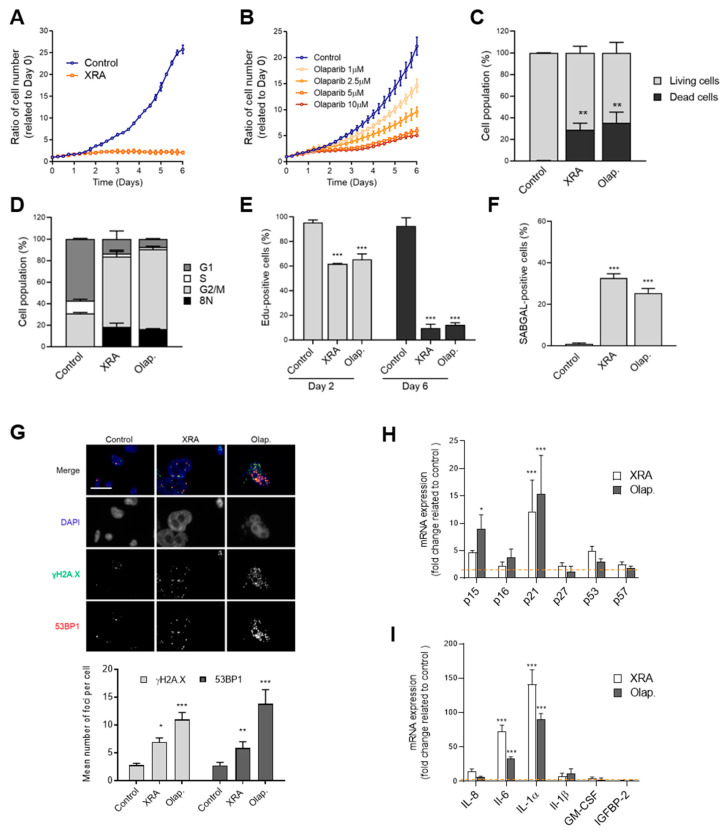
Therapy-induced DNA damage triggers mixed cell fates in PC-3 cells (**A**,**B**) Cell proliferation curves of PC-3 expressing H2B-GFP and (**A**) irradiated with 8 Gy of X-rays (XRA) or (**B**) treated with increasing concentrations of olaparib (Olap). (**C**–**I**) PC-3 cells were irradiated with 8 Gy XRA or treated with 5 µM Olap. Data are the mean ± SD of triplicate and are representatives of three independent experiments. (**C**) The cumulative cell death of PC-3 was analyzed by flow cytometry 6 days after XRA or Olap treatment. (**D**) Flow cytometry analysis of cell cycle populations following 6 days of exposure of PC-3 cells to XRA or Olap. (**E**) Cell proliferation was assessed by incorporation of EdU for 24 h at 1 or 5 days after XRA or Olap treatment. (**F**) SA-β-gal assay was performed on PC-3 cells 6 days after XRA or Olap treatment. (**G**) Representative images (top) and quantifications (bottom) of γH2AX (green) and 53BP1 (red) foci per nucleus in PC-3 cells 6 days following XRA or Olap treatment. The merged red and green channels show colocalization in yellow and DAPI is shown in blue. Scale bar = 20 µM (**H**,**I**) Relative mRNA levels of (**H**) CDKi or (**I**) SASP factors were evaluated by real-time Q-PCR in PC-3 cells irradiated or treated with Olap for 6 days. The values represent the fold change expression related to non-treated controls (represented by the dashed line). (**C**–**I**) The mean ± SD of three independent experiments is shown. Statistical analysis: (**C**,**E**,**F**,**G**) two-tailed Student’s *t*-test; (**H**,**I**) two-way ANOVA. * *p* < 0.05, ** *p* < 0.01, *** *p* < 0.001.

**Figure 3 cells-09-01593-f003:**
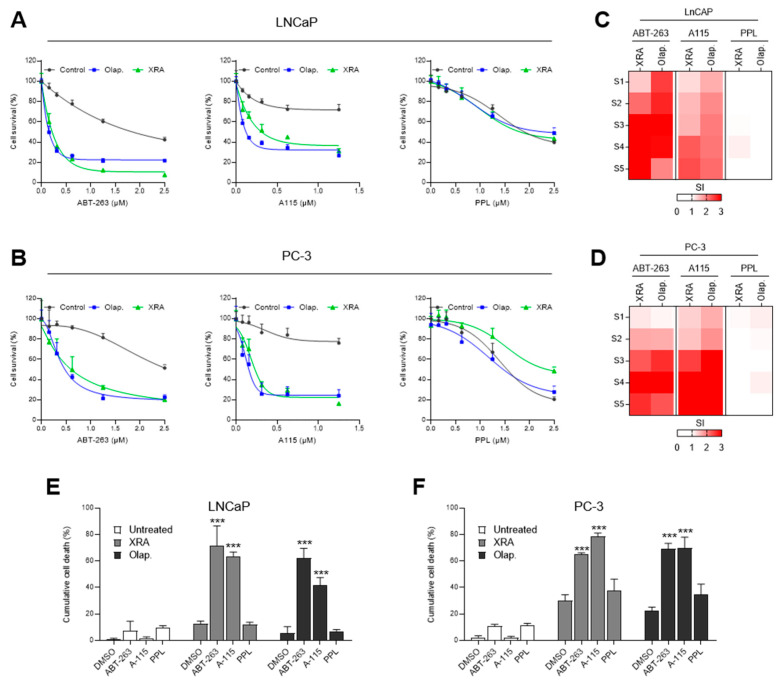
Bcl-2 family senolytics synergize with irradiation and olaparib. (**A**–**D**) LNCaP and PC-3 were infected with lentiviruses expressing H2B-GFP and treated with 8 Gy of X-Rays (XRA) or 5 µM olaparib (Olap) for 6 days, alone or in combination with increasing concentrations of ABT-263 (left), A-1155463 (A-115; middle) or PPL (right). Cell survival curves of LNCaP (**A**) or PC-3 (**B**) for the different combination treatments. Heat maps of senolytic indexes (SI) for LNCaP (**C**) or PC-3 (**D**) treated with the indicated combination treatments. S1 to S5 correspond to increasing senolytic concentrations (see Appendix A). (**E**,**F**) The cumulative cell death of (**E**) LNCaP and (**F**) PC-3 were analyzed by flow cytometry 6 days after 8 Gy XRA or following 6 days’ exposure to 5 μM Olap, alone or in combination with 0.625 µM ABT-263, 0.3125 µM A-115 or 0.625 µM PPL. (**A**,**B**) Data are the mean ± SD of triplicate and are representatives of three independent experiments. (**C**–**F**) The mean ± SD of three independent experiments is shown. Data were analyzed using the two-tailed Student’s t-test. *** *p* < 0.001.

**Figure 4 cells-09-01593-f004:**
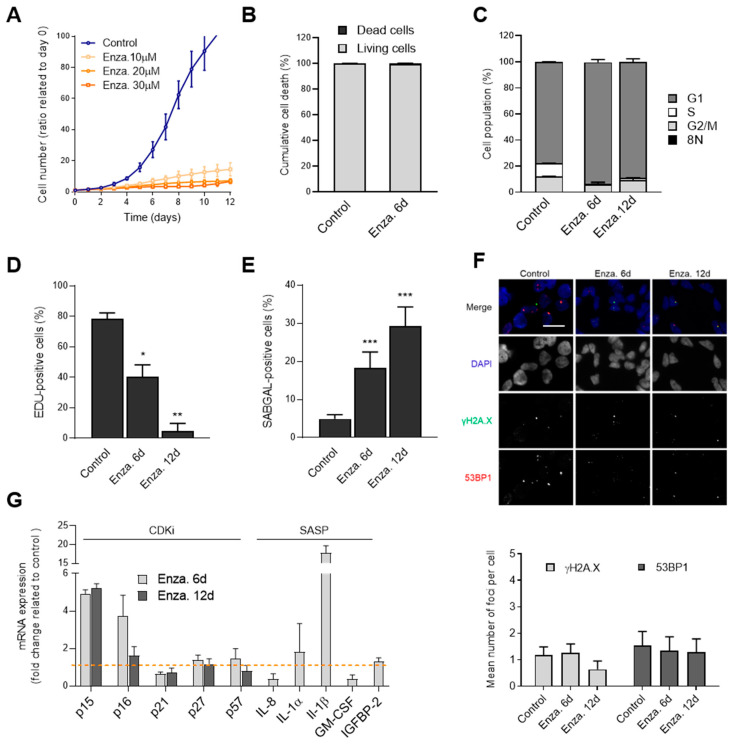
Enzalutamide triggers a senescence-like state in LNCaP cells. (**A**) Cell proliferation curves of LNCaP expressing H2B-GFP and treated with increasing concentrations of enzalutamide (Enza). Data are the mean ± SD of triplicate and are representatives of three independent experiments. (**B**) The cumulative cell death of LNCaP was analyzed by flow cytometry 6 days after 10 µM Enza exposure. (**C**) Flow cytometry analysis of LNCaP cell cycle populations following 6 or 12 days’ exposure to 10 µM Enza. (**D**) Cell proliferation was assessed by incorporation of EdU for 24 h at 5 or 11 days after 10 µM Enza exposure. (**E**) SA-β-gal assay was performed 6 or 12 days after 10 µM Enza exposure. (**F**) Representative images (top) and quantifications (bottom) of γH2AX (green) and 53BP1 (red) foci per nucleus in LNCaP cells 6 days following 6 or 12 days exposure to 10 µM Enza. The merged red and green channels show colocalization in yellow and DAPI is shown in blue. Scale bar = 20 µM (**G**) Relative mRNA levels of CDKi and SASP factors were evaluated by real-time Q-PCR in LNCaP cells exposed to 10 µM Enza for 6 or 12 days. The values represent the fold change expression related to non-treated controls (represented by dashed line). (**B**–**F**) The mean ± SD of three independent experiments is shown. Statistical analysis: (**B**–**F**) two-tailed Student’s *t*-test; (**G**) two-way ANOVA. * *p* < 0.05, ** *p* < 0.01, *** *p* < 0.001.

**Figure 5 cells-09-01593-f005:**
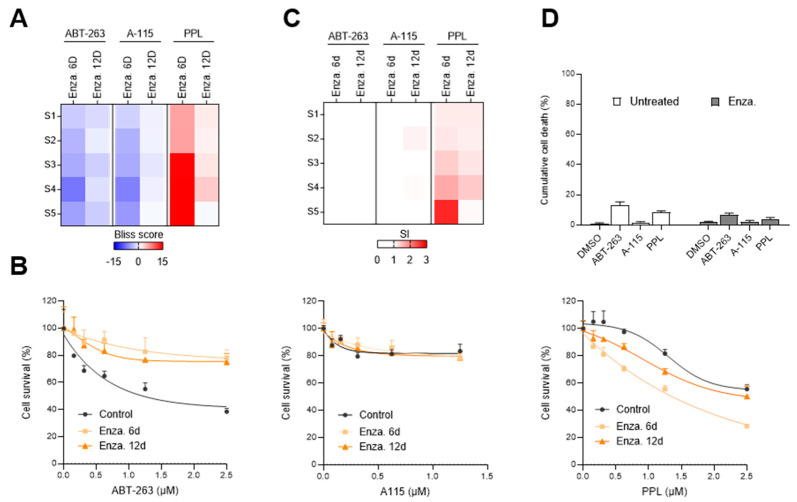
Enza-TIS cells are not sensitive to senolytics. (**A**–**D**) LNCaP expressing H2B-GFP were treated with 10 µM enzalutamide (Enza) for 6 or 12 days, alone or in combination with increasing concentrations of ABT-263 (left), A-1155463 (A-115, middle) or PPL (right). (**A**,**C**) Heat maps of bliss scores (**A**) and senolytic indexes (SI) (**C**) for LNCaP treated with the indicated combination treatments. S1 to S5 correspond to increasing senolytic concentrations (see Appendix A). (**B**) Cell survival curves of LNCaP expressing H2B-GFP and treated with the indicated combination treatments. (**D**) The cumulative cell death of LNCaP was analyzed by flow cytometry 6 days after 10 μM Enza exposure, alone or in combination with 0.625 µM ABT-263, 0.3125 µM A-115 or 0.625 µM PPL (**A**,**C**,**D**). The mean ± SD of three independent experiments is shown. (**B**) Data are the mean ± SD of triplicate and are representatives of three independent experiments. Data were analyzed using the two-tailed Student’s *t*-test.

**Figure 6 cells-09-01593-f006:**
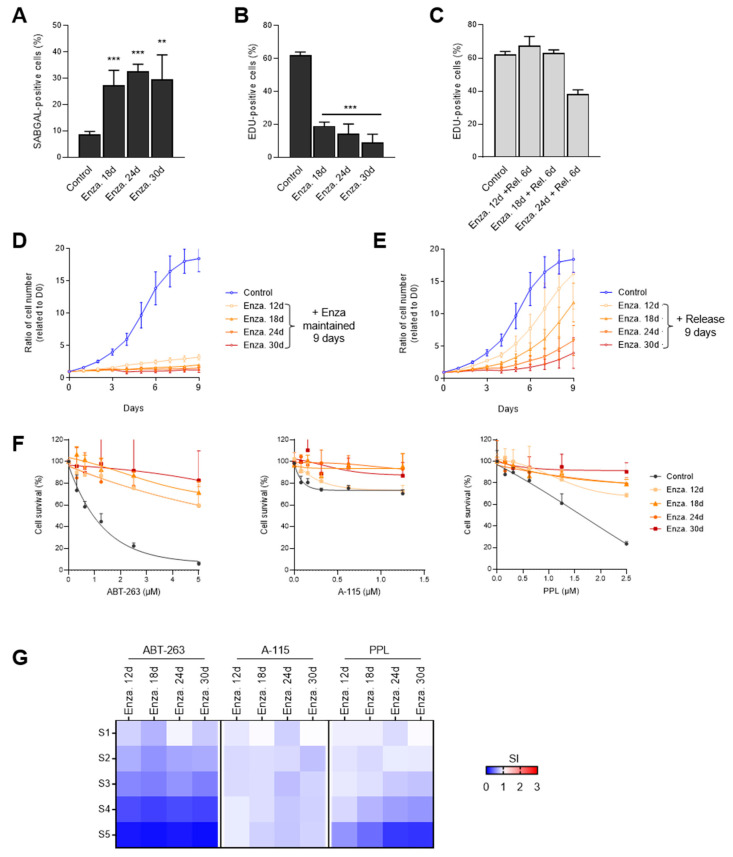
Enza-TIS is reversible. (**A**) SA-β-gal assay and (**B**) quantification of EdU-positive cells were performed in LNCaP cells following 18, 24 or 30 days of enzalutamide (Enza) treatment. (**C**) EdU staining was performed in LNCaP for which 10 µM Enza was released during the last 6 days of long-term Enza treatment. Cells were pulsed 24 h before the staining. (**A**–**C**) The mean ± SD of three independent experiments is shown (**D**–**E**) LNCaP cells expressing H2B-GFP were exposed to 10 µM Enza for 12, 18, 24 or 30 days and cell proliferation curves were determined for 9 days while Enza was (**D**) maintained or (**E**) released. Data are the mean ± SD of triplicate and are representatives of three independent experiments. (**F**) LNCaP expressing H2B-GFP were exposed to 10 µM Enza for 12, 18, 24 or 30 days, and then cell survival curves were determined during a 6-day exposure to Enza, alone or in combination with increasing concentrations of ABT-263 (left), A-1155463 (A-115, middle) or PPL (right). Data are the mean ± SD of triplicate and are representatives of three independent experiments. (**G**) Heat maps of senolytic indexes (SI) for LNCaP treated with the indicated combination treatments. S0 to S5 correspond to increasing senolytic concentrations (see Appendix A). The mean ± SD of three independent experiments is shown. Data were analyzed using the two-tailed Student’s *t*-test. ** *p* < 0.01, and *** *p* < 0.001.

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
