# Peer review of "DNA Damage- But Not Enzalutamide-Induced Senescence in Prostate Cancer Promotes Senolytic Bcl-xL Inhibitor Sensitivity"

_cells, 2020, doi:10.3390/cells9071593_

Round 1

Reviewer 1 Report

In this paper, Malaquin et al investigate the combinatorial effects of senolytic drugs and anti cancer treatments in prostate cancer cell lines.

The paper is well written, the focus of the study is clear and the research is sound. All the data are well presented and explained and the flow of the experiments is easy to follow. Validation of the findings on primary tumor cells would have been more of impact for the research, but I understand it could be not always feasible.

I just have a few minor comments:

Figure S1E: could you provide other b-gal images?  these pictures are not representative of the quantification, it is not even possible to see the blue signal in treated cells.

regarding the SASP, from the gene expression analysis it seems that there is no a major activation of the program in both conditions (irradiation and drug). However, the cytokine array in figure S1F show induction of some cytokines after irradiation. It would be nice to perform the cytokine analysis secretion also for the treated cells. This apply also for Figure 2 with treatments on PC3 cells.

Enza treatment on Figure 4: How did you measure cell death?  it seems that the treatment induce cell cycle arrest, all the cells are alive- no DNA damage -

From the graphs, it seems that only a % of cells are senescent and some other are Edu positive after tretament. How about the other remaining cells that are still alive? Could you discuss this point?

S5E The blue signal of beta-gal is not visible

Line 363 you stated "We did not observe any significant changes 363 in the number of 53BP1 and γH2AX foci between the different time points, which suggests that the phenotypes observed at 12 days did not evolve over time":

This is not completely in line with images provided: in Figure S5FS5F it seems that DNA damage foci are bigger in treated cells than controls. Could you quantify the number of 53BP1/gH2AX foci positive cells? could you quantify the intensity of DNA damage foci?

Reviewer 2 Report

The authors show that a spectrum of PCa senescence phenotypes can be induced by clinically relevant therapies. The authors found that DNA damage inducers like irradiation and poly (ADP-ribose) polymerase1 inhibitors triggered a stable PCa-TIS independent of the p53 status. On the other hand, enzalutamide triggered a reversible senescence-like state that lacked evidence of cell death or DNA damage. Using a small senolytic drug panel authors found that senescence inducers dictated senolytic sensitivity. While Bcl-2 family anti-apoptotic inhibitor were lethal for PCa-TIS cells harboring evidence of DNA damage, they were ineffective against enzalutamide-TIS cells. Interestingly, piperlongumine, which was described as a senolytic, acted to enhance enzalutamide-TIS without promoting cell death. These results suggest that TIS phenotypic hallmarks need to be evaluated in a context-dependent manner because they can vary with senescence inducers, even within identical cancer cell populations. This is an interesting and important study proposing novel therapy for metastatic castrate-resistant PCa targeting senescent cancer cells. Therefore, this reviewer feels that this study is appropriate for publication after minor revision.

Minor point:

Comment 1: In line 238, authors describe that most of cells were SA-b-gal positive. But, in Figure 2F, actually about 30% cells were SA-b-gal positive cells. This reviewer thinks that this is inconsistent. So, authors should rewrite the explanation of Figure 2F.

Comment 2: In Figure 1 and 2, authors showed expression of SASP factors for one of senescent cell markers. But, in Figure 4, there is no explanation and exhibition about SASP factors in Enza-treated cells. Are there any reasons for this? This reviewer thinks that if possible, authors should show expression of SASP factors in Enza-treated cells.

Comment 3: In lines 333-335, explanation of Figure 5B is appeared before Figure 5A. This reviewer thinks that order of Figure and explanation should be modified correctly.
